# Developmental changes and metabolic reprogramming during establishment of infection and progression of *Trypanosoma brucei brucei* through its insect host

Arunasalam Naguleswaran[1], Paula Fernandes[1,2], Shubha Bevkal[1,2¤], Ruth Rehmann[1], Pamela Nicholson[3], Isabel Roditi[1]*

**1** Institute of Cell Biology, University of Bern, Bern, Switzerland, **2** Graduate School of Cellular and Biomedical Sciences, University of Bern, Bern, Switzerland, **3** Next Generation Sequencing Platform, University of Bern, Bern, Switzerland

¤ Current address: Department of Microbiology and Immunology, Weill Cornell Medical College, New York, New York, United States of America
* isabel.roditi@izb.unibe.ch

**Data Availability Statement:** Raw read files are deposited at the European Nucleotide Archives

## Abstract

*Trypanosoma brucei ssp.*, unicellular parasites causing human and animal trypanosomiasis, are transmitted between mammals by tsetse flies. Periodic changes in variant surface glycoproteins (VSG), which form the parasite coat in the mammal, allow them to evade the host immune response. Different isolates of *T. brucei* show heterogeneity in their repertoires of VSG genes and have single nucleotide polymorphisms and indels that can impact on genome editing. *T. brucei brucei* EATRO1125 (AnTaR1 serodeme) is an isolate that is used increasingly often because it is pleomorphic in mammals and fly transmissible, two characteristics that have been lost by the most commonly used laboratory stocks. We present a genome assembly of EATRO1125, including contigs for the intermediate chromosomes and minichromosomes that serve as repositories of VSG genes. In addition, *de novo* transcriptome assemblies were performed using Illumina sequences from tsetse-derived trypanosomes. Reads of 150 bases enabled closely related members of multigene families to be discriminated. This revealed that the transcriptome of midgut-derived parasites is dynamic, starting with the expression of high affinity hexose transporters and glycolytic enzymes and then switching to proline uptake and catabolism. These changes resemble the transition from early to late procyclic forms in culture. Further metabolic reprogramming, including upregulation of tricarboxylic acid cycle enzymes, occurs in the proventriculus. Many transcripts upregulated in the salivary glands encode surface proteins, among them 7 metacyclic VSGs, multiple BARPs and GCS1/HAP2, a marker for gametes. A novel family of transmembrane proteins, containing polythreonine stretches that are predicted to be O-glycosylation sites, was also identified. Finally, RNA-Seq data were used to create an optimised annotation file with 5' and 3' untranslated regions accurately mapped for 9302 genes. We anticipate that this will be of use in identifying transcripts obtained by single cell sequencing technologies.

(ENA) http://www.ebi.ac.uk/ena under study
PRJEB44373 for genome sequencing and
PRJEB44374 for transcriptome sequencing. The
assembled megabase chromosomes and contigs
corresponding to the intermediate and
minichromosomes are deposited at NCBI
bioproject (http://www.ncbi.nlm.nih.gov/bioproject)
under study PRJNA723622. All contigs are
deposited at https://boris.unibe.ch/156137/.

**Funding:** This research was funded by University
of Berne and grants from the Swiss National
Science Foundation (no. 310030_184669) and
HHMI Senior International Scholars Program (no.
55007650) to IR. The funders had no role in study
design, data collection and analysis, decision to
publish, or preparation of the manuscript.

**Competing interests:** The authors have declared
that no competing interests exist.

## Author summary

*Trypanosoma brucei ssp.* are single-celled parasites that cause two tropical diseases: sleeping sickness in humans and nagana in domestic animals. Parasites survive in the host bloodstream because they periodically change their surface coats and also because they can switch from slender dividing forms to stumpy non-dividing forms. The latter can be transmitted to their second host, the tsetse fly. Although closely related, different geographical isolates differ in their repertoire of surface coats and have small, but important differences in their DNA sequences. In addition, laboratory strains that are transferred between mammals by needle passage lose the ability to produce stumpy forms and to infect flies. The isolate *T. b. brucei* EATRO1125 is often used for research as it produces stumpy forms and is fly transmissible. We provide an assembly of the genome of this isolate, including part of the repertoire of coat proteins, and a detailed analysis of the genes that the parasites express as they establish infection and progress through the fly. This has provided new insights into trypanosome biology. The combined genomic (DNA) and transcriptomic (RNA) data will be useful resources for the trypanosome research community.

## Introduction

*Trypanosoma brucei* ssp., the causative agents of human sleeping sickness and nagana in cattle in sub-Saharan Africa, are unicellular organisms that are transmitted between mammals and tsetse flies. During their life cycle, they undergo several differentiation steps to ensure population expansion and transmission between their two hosts. In the mammalian host there are two forms: actively dividing slender forms and quiescent stumpy forms that can establish fly infections upon transmission. Long believed to be restricted to the bloodstream and the central nervous system, it has now been shown that trypanosomes are also present in adipose tissues and skin [1,2]. In the tsetse fly, trypanosomes undergo multiple rounds of differentiation as they migrate through different compartments of the alimentary tract. A limited number of life-cycle stages have been identified based on their morphology: procyclic and mesocyclic forms in the midgut, long and short epimastigotes in the proventriculus and attached epimastigotes, gamete-like forms, pre-metacyclic trypomastigotes and metacyclic trypomastigotes in the salivary glands [3–5]. Previous studies suggested that procyclic forms exist as early and late procyclic forms that are morphologically indistinguishable, but express different surface proteins. Early procyclic forms express both GPEET and EP procyclin, while late procyclic forms express only EP procyclin [6–9]. More recent studies performed with cultured early and late procyclic forms showed that more than seventy transcripts were differentially expressed between the two forms [10]. Analysis of tsetse-derived midgut forms also indicated that some of these transcripts were differentially expressed on day 3 and day 12 post infection [11]. It has been documented that trypanosomes migrating towards the proventriculus start to elongate in the anterior midgut and that this continues further in the proventriculus [9,12]. All these studies indicate that midgut forms are dynamic in nature.

During transmission the parasites are exposed to different environments and need to regulate their gene expression to adapt to new niches. Trypanosomes live in a glucose-rich milieu in the mammalian host, but once they enter the tsetse fly glucose becomes limited and they switch their metabolism to utilise amino acids, mainly proline, as a source of energy [13]. In addition, trypanosomes in the insect midgut experience increases in pH [14] and changes in

osmolarity [15] compared to the bloodstream and tissue fluids of the mammalian host. It has been shown that procyclic forms can sense and adapt to nutrient availability in culture [16]. In addition, trypanosomes deprived of glucose or amino acids react by up-regulating transcripts for amino acid transporters [17].

In previous studies of the transcriptomes of tsetse-derived trypanosomes midgut, proventricular and salivary gland forms were sampled at a single time point at 40 days post infection [18,19]. These analyses would have missed genes specifically required for the establishment of midgut infections. One of the goals of the current study has been to monitor changes in the parasite transcriptome *in vivo*, as a function of time and tissue, and to correlate this with the biology of trypanosome development in the tsetse host. For these experiments we used *Trypanosoma brucei brucei* EATRO1125 (AnTAR1 serodeme) [20], an isolate which is favoured by many laboratories because it is pleomorphic, giving rise to both slender and stumpy forms in its mammalian host, and fly transmissible. To date, however, the genome sequence for EATRO1125 is not publicly available, so information has to be inferred from the reference strain TREU927. Here we provide a genome assembly with annotations and the *de novo* assembled transcriptome of the tsetse life-cycle stages. This is the first comprehensive transcriptome obtained for *T. b. brucei* EATRO1125 from the midgut, proventriculus and salivary glands of its tsetse host *Glossina morsitans morsitans*. Reads of 150bp enabled us to distinguish closely related genes that are differentially regulated through the life cycle. In addition, we identified the repertoire of metacyclic variant surface glycoproteins (mVSGs) and used the 5' and 3' untranslated regions from full length cDNAs to create an optimised annotation file. This can be used for a variety of next generation sequencing approaches, such as mapping transcripts obtained in single cell RNA-seq.

## Methods

### Parasite cultivation

Fly transmissible bloodstream forms of *Trypanosoma brucei brucei* EATRO 1125, expressing VSG AnTat 1.1 [20], were cultivated in HMI-9 supplemented with 1.1% methylcellulose and 10% heat inactivated foetal bovine serum (FBS) [21]. Slender bloodstream forms differentiated into stumpy forms upon reaching a density of $5 \times 10^6$ ml$^{-1}$ [22]; these were used for tsetse infections.

### Genomic DNA isolation and PacBio long-read HiFi sequencing

DNA quality control tests, library preparations and sequencing runs were performed at the Next Generation Sequencing Platform, University of Bern, Switzerland and the Genomics and Proteomics Core Facility, German Cancer Centre, Heidelberg, Germany. Genomic DNA was isolated from procyclic forms as described [23]. Prior to SMRTbell library preparation, the genomic DNA was further purified using a Genomic DNA Clean & Concentrator clean up kit (D4011, Zymo Research) and then assessed for quantity, quality and purity using a Qubit 4.0 fluorometer (Qubit dsDNA HS Assay kit; Q32851, Thermo Fisher Scientific), an Advanced Analytical FEMTO Pulse instrument (Genomic DNA 165 kb Kit; FP-1002-0275, Agilent) and a deNovix DA-11 Series, full-spectrum, UV-Vis spectrophotometer, respectively. SMRTbell libraries for sequencing were prepared using the SMRTbell Express Template Prep Kit 2.0: Procedure & Checklist, PN 101-853-100, version 03. The only exception to this protocol was that the gDNA was not sheared using the Megaruptor 3 system, but rather using the Megaruptor 2 system (Diagenode) with the software settings at 20Kb and following their user manual, version 02, 03.2018. The sheared DNA was subjected to ExoVII treatment to remove single-stranded ends, followed by DNA damage repair and DNA end repair treatments, an A-tailing

step and adapter ligation. The DNA was then purified using 1X AMPure beads prior to exonuclease treatment to remove excess ligation products. Prior to size selection, the SMRTbell templates underwent another 1X AMPure bead-based clean up. Size selection was performed using a BluePippin System (BLU0001, Sage Science) as outlined in the protocol and followed by a 1X AMPure PB Bead purification step. The concentration and size of the resulting SMRTbell library was assessed as described in the QC steps above. The final library insert size was 19,834 bp.

PacBio Sequencing primer v2 was annealed and bound to the polymerase using Binding Kit v3.0, with a 4 h binding time at 30˚C. The binding complex was cleaned using a 1.2X AMPure PB bead-based clean-up. The libraries were loaded at an on-plate concentration of 8 pM using diffusion loading, along with the use of a Spike-In internal control 3.0. SMRT sequencing was performed in CCS mode on the Sequel System with a Sequel Sequencing Plate 3.0, 7.5 h pre-extension, 20 h movie time and with PacBio SMRT Link v8.0.

### *De novo* assembly using PacBio HiFi reads

HiFi/circular consensus sequencing (CCS) reads were used for *de novo* assembly with Canu assembler 2.1 with default parameters [24]. Contigs generated in this fashion were submitted to the Companion server (http://companion.gla.ac.uk/) [25], using default settings and guided by transcriptome data, to generate pseudo chromosomes corresponding to the 11 megabase chromosomes of *T. b. brucei* TREU927. A general feature format (GFF) file was created by transferring the genome annotations using the Liftoff tool [26].

### Infection and maintenance of tsetse flies

*Glossina m. morsitans* pupae were obtained from the Department of Entomology, Slovak Academy of Science (Bratislava). Tsetse flies were infected according to standard procedures described previously [27,28]. Pupae and flies were maintained at 27˚C, 80% humidity on a 12 h/12 h light/dark cycle. Culture derived stumpy forms generated in HMI-9 medium supplemented with 1.1% methylcellulose were diluted 1:5 in HMI-9 with 10% FBS and centrifuged for 10 min at 1200g (2400 rpm) to remove the methylcellulose. Parasites were resuspended in defibrinated horse blood obtained from TCS Biosciences Ltd (Buckingham, UK; cat no: HB035) at a density of $2 \times 10^6$ ml$^{-1}$ and used for fly infection. Teneral flies were offered an infectious blood meal 24 h and 48 h after emergence, and subsequently fed with defibrinated horse blood at 48–60 h intervals.

Flies were dissected at days 3, 7, 11, 15 and 28 post infection to isolate trypanosomes from tsetse midgut samples. In addition, trypanosomes were isolated from the proventriculus and salivary glands on day 28. Isolated trypanosomes were immediately lysed in solution D [29] and stored at -70˚C until total RNA isolation was performed.

### RNA isolation and RNA-Seq analysis

RNA-Seq was performed using 3 biological replicates for each tissue and time point. Replicate 1 used cultured bloodstream forms derived from a mouse infection [21]. Replicates 2 and 3 used cultured bloodstream forms derived after tsetse transmission of the bloodstream forms used in replicate 1. These were adapted directly to culture after isolation from the fly salivary glands [22]. Flies were always dissected 2 days after their last blood meal in order to minimise perturbations caused by nutritional status.

Total RNA was isolated as described previously [30] and subjected to DNase treatment to remove residual genomic DNA contamination. Library preparation and sequencing were performed at Fasteris, Geneva, Switzerland. Illumina cDNA libraries were prepared using a

TruSeq Stranded mRNA Library Prep kit. Sequencing of cDNA libraries was performed using Illumina Nextseq sequencing systems with 150 bp read lengths. Reads were mapped to the *T. b. brucei* 927 reference genome version 5, using the bowtie2 tool available in the Galaxy Interface (usegalaxy.org) with default parameters. Mapping to the genome was used to visualise the data on Gbrowse and to estimate transcript abundance. The featureCounts tool [31] available in the Galaxy platform was used to extract read counts using GFF annotation file TritrypDB-44 version and RPM values were calculated. Bioconductor package DESeq2 [32], using a cut-off of >10 mean counts, was used to identify differentially expressed genes from biological triplicates. The principal component analysis plot was generated using DESeq2 and ggplot2 packages in R, based on the previously transformed log2 raw count matrix [32,33]. Heatmaps were created by using the heatmap.2 function from the gplots package [34]. Data were z-score normalised and all heatmaps were based on the complete agglomeration method for clustering gene expression across the samples.

### *De novo* transcriptome assembly and generation of an optimised annotated file

*De novo* assembly of transcripts from tsetse forms was performed using Trinity-v2.5.1 [35] with default parameters. To exclude RNA contamination from tsetse or symbiotic bacteria, only reads that mapped to EATRO1125 pseudo chromosomes were used. Transcripts of midgut, proventricular and salivary gland forms were assembled separately. A general transfer format (GTF) file for the TREU927 genome was reannotated using the transcriptome of tsetse-derived trypanosomes. To achieve this, we used RNA-seq reads that mapped to EATRO1125 pseudo chromosomes and remapped them to the TREU927 genome. The BAM file obtained by mapping was used as an input file for Stringtie to create a GTF file with the following parameters (input.bam -o output.gtf—rf -i). The preliminary GTF file obtained from Stringtie was further annotated with the Gffcompare tool to transfer the TREU927 gene IDs. Missing information was incorporated manually after inspection of RNA-seq coverage. Metacyclic VSG transcripts were extracted from de novo assembled salivary gland transcripts using the unique 16mer sequence (GTGTTAAAATA-TATCA) found in all VSGs [36].

## Results

### Genome and transcriptome assembly of *T. b. brucei* EATRO1125

Genomic DNA from *T. b. brucei* EATRO1125 was subjected to PacBio sequencing. We obtained 321,316 HiFi/CCS reads (1.35 x $10^9$ bases). This amounted to 37.4-fold coverage for the 11 megabase chromosomes; extended to the mini- and intermediate chromosomes, coverage was 21-fold. The assembly resulted in 842 contigs, varying in length from 3.3 Mb to 1262 bp, with a total length of 64.1 Mb including megabase, intermediate-sized and minichromosomes [37,38]. The number of reads for each contig and basic assembly statistics are provided in S1 Table. These contigs were subsequently scaffolded into 11 chromosomes using the Companion server for parasite genome annotation (http://companion.gla.ac.uk/) [25]. Chromosomes 1–11 correspond to the megabase chromosomes in TREU927, but are longer (36Mb compared to 32.9 Mb). S1 File shows that insertions occur on all megabase chromosomes; these are largely non-coding. The remaining 685 contigs (28.1 Mb in total) could not be assigned to the megabase chromosomes. Of these, 73 contigs contained 177 bp repeats, which are characteristic of mini- and intermediate chromosomes [38], and 98 contained telomeric repeats. Many of these contigs were close to the expected sizes of these categories of

chromosomes. Further analysis revealed that the majority of them encoded a single VSG at the end of the contig; few encoded more than one VSG.

Companion, which uses Augustus to annotate genes, identified 13355 coding genes and 272 non-coding genes. Comparison of the EATRO1125 and TREU927 genomes revealed that the two stocks share 7465 orthologs. Furthermore, 730 paralogs and 736 singletons were identified in EATRO1125, while TREU927 had 528 paralogs and 435 singletons. The majority of the singletons and paralogs in EATRO1125 encoded hypothetical proteins or putative VSGs (S1 File). The Liftoff tool [26] was used to transfer TREU927 annotations to EATRO1125 and create a GFF3 file (S2 File). Liftoff identified 10899 genes in TREU927 and transferred the gene features to the EATRO1125 GFF3 file. Unbiased assessment of the genome assembly and annotation was performed for EATRO1125 and TREU927 with BUSCO version 5.0.0 using eukaryota_odb10. The analysis produced similar percentages of complete (C), complete and single-copy (S), complete and duplicated (D), fragmented (F), and missing (M) genes and proteins for both isolates (S1 Table). To validate our genome assembly, we examined the procyclin loci, which are known to differ between TREU927, Lister 427 and EATRO1125 [7,39,40]. The assembly of chromosome 6 of EATRO1125 is identical to previously published data with respect to the number of GPEET repeats and single copies of GRESAG2 and PAG3 [39]. In addition, the assembly of chromosome 10 confirmed the previously described organisation of EP1 and EP2 procyclin and downstream procyclin associated genes (PAGs) 1, 2, 4 and 5 on one copy of the chromosome and a PAG1/2 fusion, and no PAG5 on the second copy [40]. Furthermore, to demonstrate the usefulness of EATRO1125 genome, we compared the mapping rates of various types of next generation sequencing data compared to the TREU927 genome. For bloodstream forms, the mapping rate for the transcriptome was 98.9% for EATRO1125 versus 94.57% for TREU927. This is largely due to the VSG repertoire. Importantly, for ChIP-seq, the rate was 95.23% for EATRO 1125 versus 77.58% for TREU927 (S1 Table). The increased ChIP-seq mapping rate could be due to better coverage of the genome, including the VSG genes, sequence polymorphisms or genome expansion in the non-coding regions.

We next used the assembled genome to extract EATRO1125-specific reads from Illumina RNA-seq performed with samples from infected tsetse. This allowed us to perform *de novo* transcript assembly free of contaminating fly RNA (S3 File) and resulted in 6200 non-redundant transcripts that started with the spliced leader: 3509 (midgut day3), 4474 (midgut day 7), 5602 (midgut day 28), 2404 (proventriculus) and 5609 (salivary gland). This information was used to annotate the 5' and 3' untranslated regions (UTRs) of 9302 genes in a GTF file (GTF; S4 File). When we compared this annotated file with the single cell RNA sequencing data generated by Vigneron and co-workers [41], there was excellent agreement between their 3' reads and the mapped ends of transcripts. Moreover, using Cellranger version 6.0.0 (10x Genomics), the percentage of reads assigned to the transcriptome increased from 41% to 48.5%, and the number of reads assigned to intergenic regions decreased from 14.5% to 4.4% compared to the GTF file generated from the current GFF file from TriTrypDB.

## Overview of RNA-seq analysis and expression of stage-specific surface markers

Culture-derived stumpy forms of *T. b. brucei* EATRO1125 were used for fly infection as previously described [22]. Flies were dissected at days (D) 3, 7, 11, 15 and 28 post infection for midgut forms and at D28 for proventricular and salivary gland forms (S1 Fig). Illumina True-seq was performed on RNA from three independent biological replicates for each time point. In order to make comparisons with previous data generated by our laboratory and others

[10,18,42–44], RNA-seq reads were mapped to the TREU 927 reference genome and read counts per transcript were extracted and used for subsequent analyses. A principal component analysis (PCA) revealed that slender, stumpy, midgut, proventricular and salivary gland forms clustered separately (Fig 1A and S2 Table). Fig 1B shows a heatmap of global expression through the life cycle. Next, we validated the RNA-seq data by analysing the expression profiles of known stage-specific genes. For completeness, data from slender and stumpy forms [10] were also included. GPEET procyclin is expressed by midgut forms during the first few days of fly infection, after which the protein is no longer detectable [7]. Our analysis showed that GPEET mRNA expression peaked at D3 and decreased sharply at all later time points in samples from the midgut, proventriculus and salivary glands (Fig 2A). EP1 procyclin, which is expressed by both early and late procyclic forms in the fly [6], was detectable in the midgut at D3 and peaked at D7 and D11. On later days its expression in the midgut was reduced, but remained high compared to that in proventricular and salivary gland forms. We also monitored the expression levels of members of the BARP family, which encode GPI-anchored proteins expressed by attached epimastigotes [45]. In this case, transcripts were not detected in the midgut during the entire time course, and were barely detected in the proventriculus on D28, but were strongly upregulated in the salivary glands (Fig 2B).

Proteins associated with differentiation (PADs) are encoded by a cluster of 8 related genes [46]. Among them, PAD1 and PAD2 are known to be specifically up-regulated in stumpy forms and PAD2 expression also responds to changes in temperature [46]. Our study confirmed that both these PADs show peak expression in stumpy forms and are not upregulated again later in the life cycle (Fig 2C). Other PADs are also differentially expressed. PAD 3, which is expressed at very low levels, is upregulated in stumpy and midgut forms, while PADs 5 and 7 show midgut-specific expression. PADs 4, 6 and 8 are up-regulated in the salivary glands, corroborating a previous study [18] (Fig 2C).

## The transcriptome of midgut forms is dynamic

When bloodstream form trypanosomes are ingested by a fly, stumpy forms differentiate into procyclic forms in the midgut. For the first few days they are found inside the peritrophic matrix. Later they migrate into the ectoperitrophic space [47]. Once they establish an infection in this compartment, they eventually migrate towards the anterior end of the midgut; during this phase they also undergo morphological changes [12,48] and invade the proventriculus. This process takes several days and is critical for successful transmission [3,47]. Since there was no information available about the trypanosome transcriptome during the establishment of midgut infection we performed a time course from D3 to D28.

When an infection is initiated with stumpy forms of EATRO1125 (AnTat1.1), the parasites differentiate and are covered by EP and GPEET procyclin within 2 days [6]. In addition to high levels of procyclin mRNAs, an analysis of the midgut D3 transcriptome showed upregulation of transcripts characteristic for procyclic forms such as cytochrome oxidase subunits [49,50] and the surface protein PSSA2 [51,52], and downregulation of bloodstream-specific transcripts such as AnTat1.1 VSG, invariant surface glycoproteins and the repressor of differentiation kinase RDK2 [10,42,53] (Fig 3A).

Previous studies performed with cultured parasites showed that early and late procyclic forms are distinct forms based on their proteomes, transcriptomes and the ability to perform group migration on semi-solid surfaces [10,11]. It was not clear, however, how this correlated with trypanosomes *in vivo*. We previously identified a number of markers that discriminated between early and late procyclic forms in culture [10,11,30]. The transcriptome of tsetse-derived trypanosomes on D3 strongly resembled that of early procyclic culture forms, most

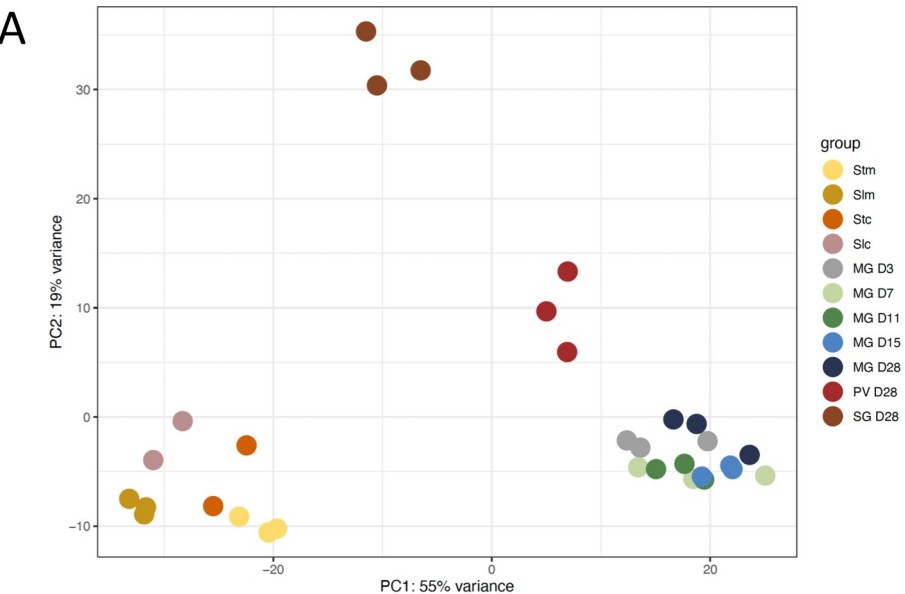

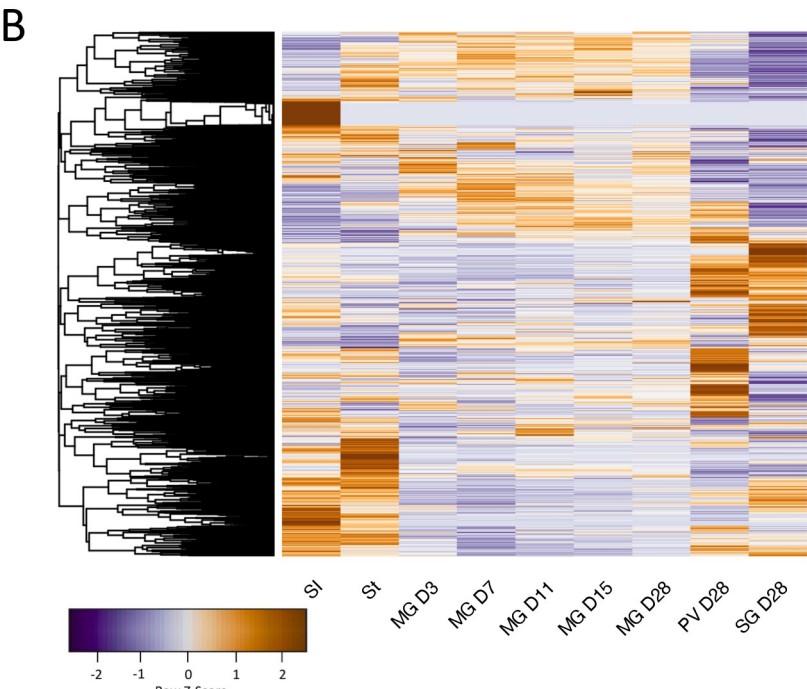

**Fig 1. Principal component analysis and heatmap of trypanosome transcriptomes from different tsetse tissues.** A: Each point represents an independent biological sample. Data for cultured slender and stumpy forms [10] and mouse-derived slender and stumpy forms [44] are included for completeness. Slc: culture-derived slender forms; Stc: culture-derived stumpy forms; Slm: mouse-derived slender forms; Stm: mouse-derived stumpy forms; MG: midgut; PV: proventriculus, SG: salivary glands. Midgut samples were collected on D3, D7, D11, D15 and D28 post infection. B: Global gene expression heatmap through the life cycle. Data were z-score normalised. Colour scale: orange, high expression; blue, low expression.

notably by the expression of GPEET, as well as the high affinity hexose transporter (THT2) family and enzymes involved in glycolysis (Fig 3). This suggests that procyclic forms scavenge and metabolise glucose at the beginning of fly infection. Other markers for early procyclic

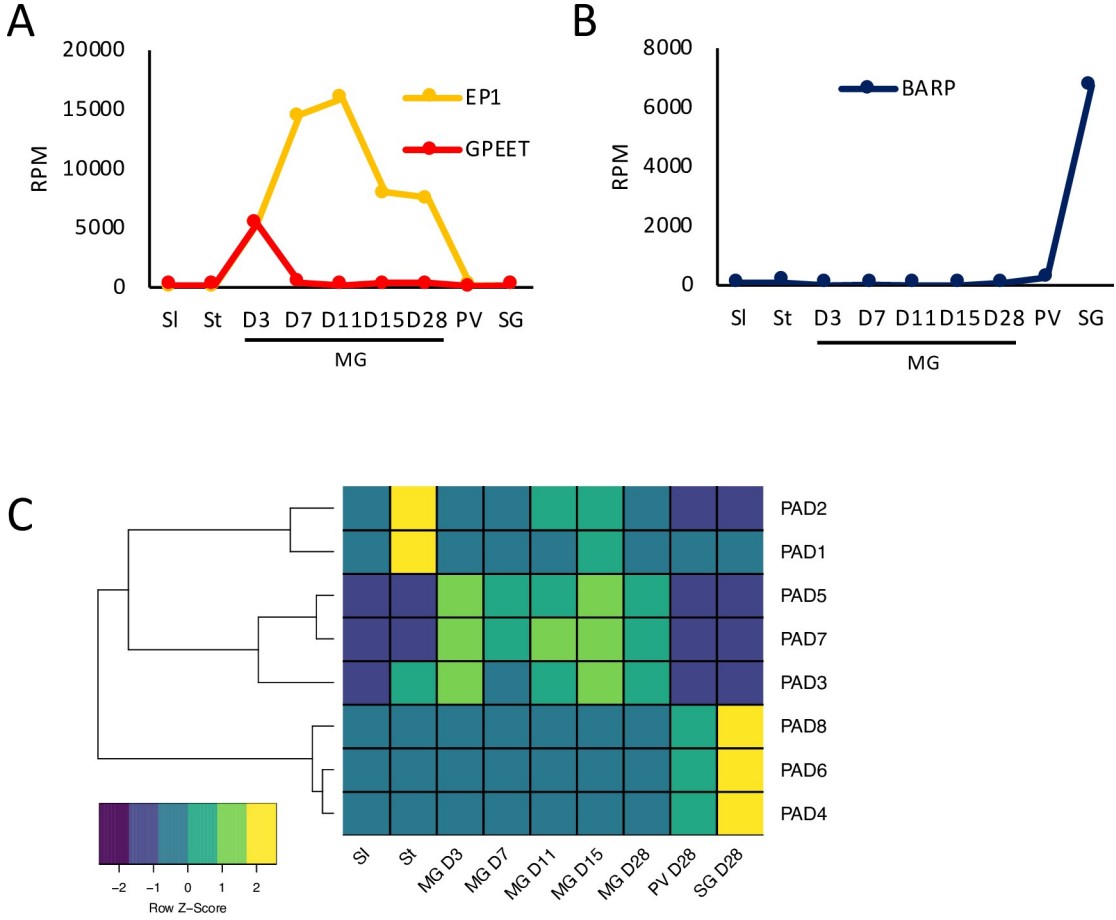

**Fig 2. Expression profiles of known stage-specific markers as a function of time and tissue.** Values are the mean of three biological replicates. A) EP and GPEET procyclin; B) BARP gene family; C) Heatmap of proteins associated with differentiation (PADs 1–8). Data for slender and stumpy forms are from [10]. Sl: slender forms; St: stumpy forms; MG: midgut; PV: proventriculus, SG: salivary glands. RPM: reads per million.

culture forms, such as putative prostaglandin synthase F, nitroreductase and calflagin Tb-44 [10,11] were also expressed at their highest levels on D3 (Fig 3 and S3 Table). Interestingly, a number of transcripts changed by D7. A total of 302 transcripts were differentially regulated between D3 and D7 midgut forms (p <0.05). Of the 118 transcripts that showed ≥2-fold regulation, 81 were up-regulated and 37 down-regulated on D7 (S3 Table). The greatest differences were observed between D3 and D15 (Fig 3B and S3 Table). These included downregulation of hexokinase 1, the glycosomal membrane protein gim5B and other markers of early procyclic forms. From D7 onwards the transcriptome stabilised and there were no significant differences in expression from D7 to D28.

Several transcripts for metabolic enzymes peaked on D7, including fructose-1,6-bisphosphatase and the tricarboxylic acid (TCA) cycle enzymes citrate synthase, isocitrate dehydrogenase and aconitase (Fig 4 and S3 Table). Transcripts for a proline/alanine transporter from the AAT7-B family (Tb927.8.7640) (17) and delta-1-pyrroline-5-carboxylate dehydrogenase also peaked on D7. These changes would be compatible with an increased need for gluconeogenesis and a switch to proline catabolism.

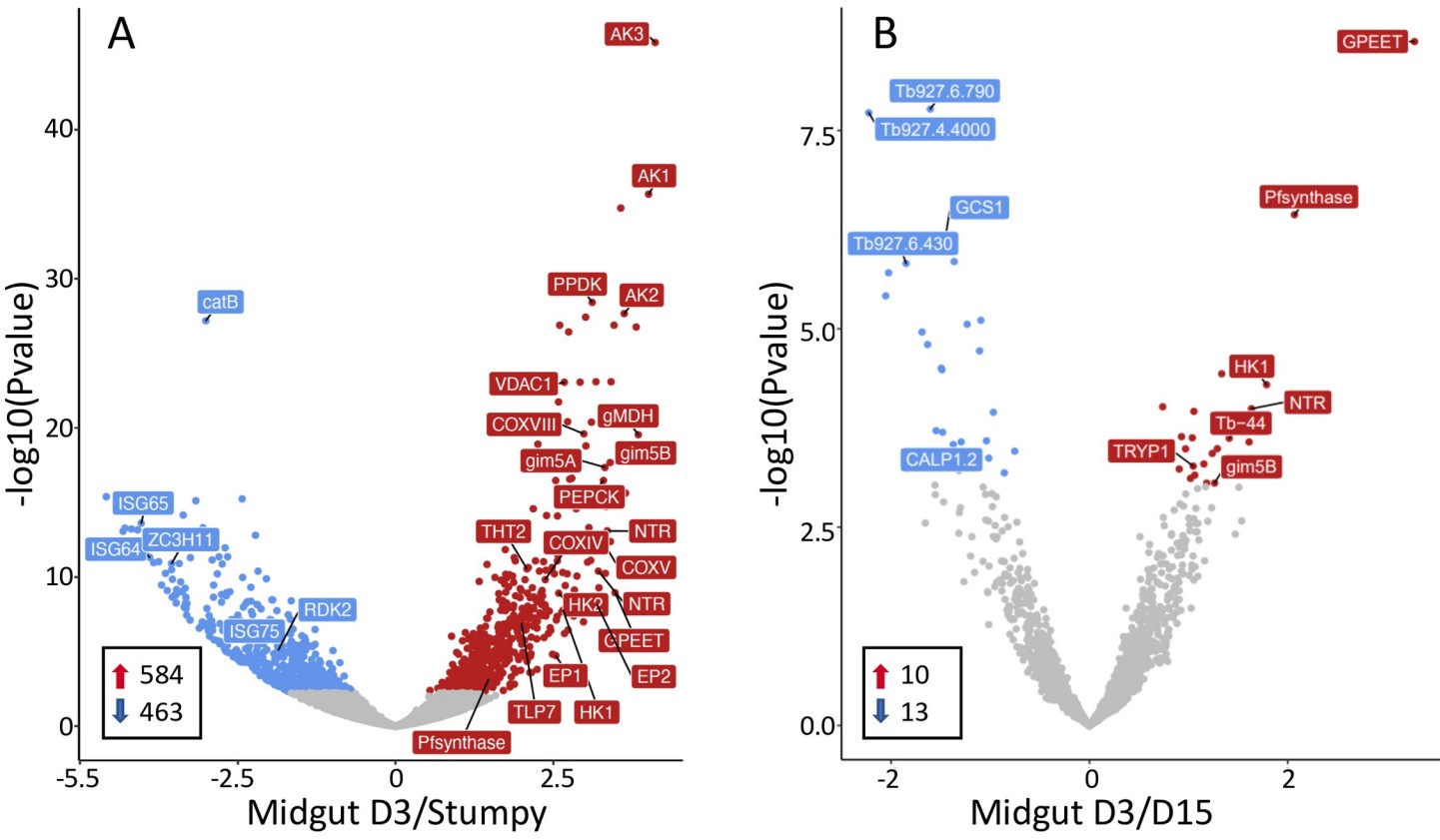

**Fig 3.** Volcano plots comparing the transcriptomes of A) culture-derived stumpy forms [10] and tsetse-derived midgut forms D3 post infection; B) midgut forms D3 versus D15. AK: arginine kinase; HK1: hexokinase 1; ISG: invariant surface glycoprotein; NTR: nitroreductase; PEPCK: phosphoenolpyruvate carboxykinase; PPDK: pyruvate phosphate dikinase; Tb-44: calflagin 44; THT2: high affinity hexose transporter. Red: Transcripts significantly upregulated on D3; blue: transcripts significantly downregulated on D3; grey: not significant. Red arrows: transcripts upregulated ≥2-fold; blue arrows, transcripts downregulated ≥2-fold.

### The transcriptome of proventricular forms

A comparison of the transcriptomes from the midgut and proventriculus on D28 revealed 1757 genes that were differentially expressed (p <0.05), 869 of them ≥2-fold. Of these, 375 transcripts were upregulated and 494 downregulated (S4 Table). Upregulated transcripts included all four core histones and linker histone H1, DNA replication licensing factors MCM5 and 6, ribonucleotide-diphosphate reductases and thymidine kinase. Gene ontology (GO) analysis also identified components of chromatin and DNA packaging as significant categories (S4 Table). This is consistent with proventricular forms completing S-phase and subsequently undergoing an asymmetric division to yield a long and a short epimastigote [9,54]. In general, however, the cells appear to be less active than midgut forms. There is a decrease in transcripts encoding translation initiation and elongation factors (EIF4A1, EIF4G4, ERF1, EEF1A). Furthermore, transcripts for the large subunit of RNA polymerase I (RPA1), and several nucleolar (Nopp44/46-1,2,3) and nuclear RNA binding proteins (TbRRM1, TRRM3, RBP11, RBP22) are down-regulated. It has been shown previously that depletion of TbRRM1 results in chromatin compaction [55]. Decreases in fatty acyl CoA synthetases (ACS1, ACS4) and fatty acid elongase (ELO1) imply that the parasites do not synthesise new membranes. In addition, threonine dehydrogenase, which is used by midgut forms to produce acetate for fatty acid synthesis [56,57], declines sharply. This would be consistent with the observation that, apart from the asymmetric division described above, trypanosomes do not replicate in the

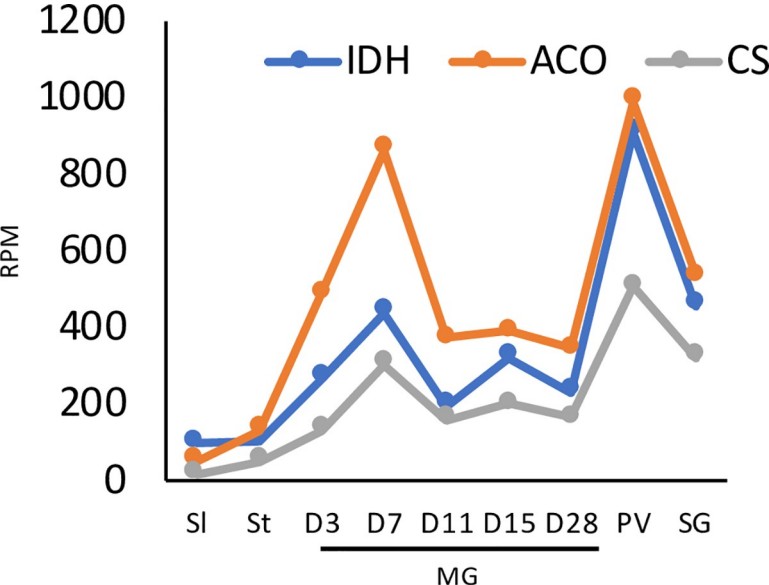

**Fig 4. Expression profiles of TCA cycle enzymes through the life cycle.** ACO, aconitase; IDH, isocitrate dehydrogenase; CS, citrate synthase. All three enzymes show bimodal expression, with highest expression at D7 in the midgut and in the proventriculus. Data for slender and stumpy forms are from [10]. RPM: reads per million.

proventriculus [9,54]). Although the proventricular population does not increase in number, the transcriptome indicates that the cells rewire their metabolic capabilities. Transcripts for components of the glycosome are reduced, while the TCA enzymes citrate synthase, isocitrate dehydrogenase and aconitase increase, surpassing their levels in the midgut on D7 (Fig 4).

The proventriculus (pH 10.6) is considerably more alkaline than the midgut when flies receive bloodmeals every 2–3 days (pH 8–8.5) [14]. A number of transcripts that are upregulated and show peak expression in this organ might be involved in stress responses. PHO85 has been implicated in environmental responses in budding yeast [58]. CYC7, which is a PHO85-like cyclin-dependent kinase according to HHpred analysis, is most highly expressed in the proventriculus, as are superoxide dismutases, NEK kinases and arginine kinases.

Transcripts for the major surface glycoproteins also change in the proventriculus. Two isoforms of EP procyclin (EP1 and EP3-2) were reduced 27-fold and 4.9-fold, respectively. This is consistent with previous observations that proventricular forms express EP, but in lower amounts than midgut forms [9,18]. Read counts for different isoforms of BARP were 5-10-fold higher than in the midgut, but still 40-fold lower than in the salivary glands (Fig 2B and S5 Table). No reads were mapped for any of the metacyclic VSGs, indicating that their transcription is stringently controlled. Based on the transcriptome, there is no evidence that other membrane proteins form a stage-specific coat in the proventriculus.

## The transcriptome of salivary gland forms

The population in the salivary glands is heterogeneous. The major life-cycle stages are epimastigotes, pre-metacyclic and metacyclic forms, but pre-meiotic stages and gametes might also be present, at least transiently [59,60]. DESeq2 revealed 1821 differentially expressed genes between proventricular and salivary gland forms (p <0.05), 684 of them upregulated ≥2-fold (S5 Table). GO analysis identified cell surface components as a significant group (S5 Table). In addition to BARPs, which are up-regulated 40-fold in the glands relative to the proventriculus, and 400-fold relative to the midgut (Fig 2B), this dataset contains invariant surface

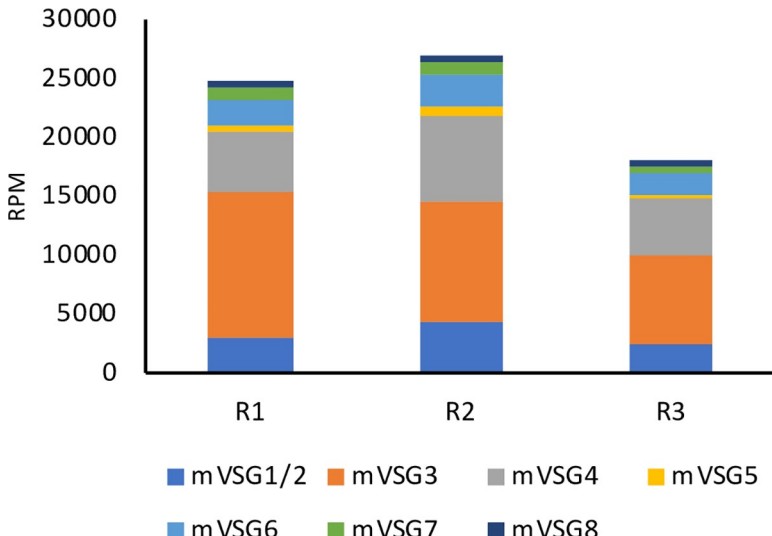

**Fig 5. Expression of metacyclic VSGs in the salivary glands.** Eight transcripts were assembled from the salivary gland transcriptome. mVSG1 and mVSG2 are alternatively processed transcripts encoding the same protein and are therefore grouped together. The same set of mVSGs was found in three biological replicates (R1–3). RPM: reads per million. Sequence information is provided in S4 File.

glycoproteins, components of the enriched surface labelled proteome [61] and putative GPI anchored proteins.

Metacyclic VSGs from EATRO1125 have not been annotated. To identify these, we took Illumina reads from the salivary glands and removed all sequences mapping to either the megabase chromosomes of TREU927 or the *G. m. morsitans* transcriptome. The remaining reads were assembled *de novo* using the Trinity tool and the transcripts were screened for the presence of the conserved 16mer in the 3' UTRs of VSGs [36]. Eight spliced and polyadenylated transcripts, encoding 7 mVSGs, were identified. mVSG1 and mVSG2 were alternatively spliced mRNAs with the same coding region; for this reason, read counts for these two mVSGs were pooled (S5 File and Fig 5). Four mVSGs corresponded to coding regions previously catalogued in the VSGome of EATRO1125 [62] (http://129.85.245.250/index.html) and one was identical to MVAT5 from *T. b. rhodesiense* [63]. Each mVSG was mapped to a contig containing telomeric repeats (S1 Table). The reads assigned to mVSGs accounted for 1.8–2.7% of the total in the salivary glands. The seven transcripts were expressed to different extents, with mVSG3 and mVSG4 predominating in each of the 3 biological replicates (Fig 5).

In addition to the mVSGs and BARPs, transcripts for several other GPI-anchored proteins were specifically upregulated in the salivary glands (Table 1). A cluster of 4 genes on chromosome 7 were recently shown to encode epimastigote-specific proteins (SGE) [41]. Three other GPI-anchored proteins, Tb927.8.950, Tb.927.8.970 and one copy of amastin (Tb927.4.3520) also show peak expression in the glands. Transcripts for another family of putative membrane proteins (based on the presence of a signal peptide and a transmembrane domain) also show maximum expression in the salivary glands. These 8 proteins, which are very similar in sequence, are characterised by a stretch of 6–18 polythreonine residues that is highly likely to be O-glycosylated (https://services.healthtech.dtu.dk/service.php?NetOGlyc-4.0). Transcripts encoding GCS1/HAP2, a membrane protein required for gamete fusion in other organisms [64–67], are also strongly up-regulated. This is compatible with the glands being the site of gamete formation [4,59,60]. Retrotransposon hotspot (RHS) proteins are a large and complex family that has been further classified into 7 subfamilies [68]. It was shown recently that RHS2,

**Table 1. Surface protein transcripts, in addition to mVSGs, specifically upregulated in the salivary glands.**

| BARPs | log$_2$FC | p-value | Other GPI-anchored proteins | log$_2$FC | p-value | Other membrane proteins | log$_2$FC | p-value |
|---|---|---|---|---|---|---|---|---|
| Tb927.9.15520 | 3.5 | 2.6E-14 | Tb927.4.3520 (amastin) | 2.0 | 1.7E-12 | Tb927.2.4760 | 2.0 | 1.1E-09 |
| Tb927.9.15530 | 4.3 | 3.1E-20 | Tb927.7.360 (SGE1) | 4.2 | 1.5E-18 | Tb927.2.4920 | 2.3 | 1.7E-08 |
| Tb927.9.15540 | 5.3 | 1.2E-25 | Tb927.7.380 | 4.5 | 2.3E-18 | Tb927.2.5290 | 2.7 | 3.9E-09 |
| Tb927.9.15550 | 4.1 | 1.8E-14 | Tb927.7.400 | 4.7 | 7.9E-26 | Tb927.2.5300 | 3.0 | 2.4E-14 |
| Tb927.9.15560 | 4.0 | 2.1E-21 | Tb927.7.420 | 3.6 | 6.8E-14 | Tb927.2.5330 | 3.7 | 2.5E-14 |
| Tb927.9.15570 | 4.9 | 3.8E-21 | Tb927.7.440 | 4.1 | 3.0E-15 | Tb927.2.5340 | 3.3 | 2.5E-14 |
| Tb927.9.15580 | 4.6 | 5.7E-27 | Tb927.8.950 | 1.5 | 5.2E-06 | Tb927.2.5350 | 3.2 | 1.2E-15 |
| Tb927.9.15590 | 2.9 | 1.7E-08 | Tb927.8.970 | 2.7 | 1.8E-15 | Tb927.2.5360 | 2.4 | 9.1E-08 |
| Tb927.9.15600 | 3.9 | 1.2E-24 | | | | Tb927.10.10770 (GCS1/HAP2) | 2.8 | 1.5E-20 |
| Tb927.9.15610 | 5.1 | 4.8E-34 | | | | | | |
| Tb927.9.15620 | 2.6 | 8.4E-08 | | | | | | |
| Tb927.9.15630 | 5.0 | 5.5E-30 | | | | | | |
| Tb927.9.15640 | 2.9 | 1.7E-12 | | | | | | |

Fold changes (FC) are relative to the proventriculus.

4 and 6 are essential, and that they are linked to transcription and RNA export in procyclic forms [69]. The complexity and interrelatedness of RHS has meant that they are usually excluded from analyses of the transcriptome and proteome, but the long reads employed in this study have enabled us to map them. Intriguingly, while the other RHS subfamilies are highly expressed in several life-cycle stages, we find that RHS7 is unique in its expression being restricted to the salivary glands (S5 Table).

As mentioned above, there are several different cell types in the salivary glands. To determine which cell types express polythreonine-containing proteins, we took advantage of the single cell dataset from salivary gland-derived parasites [41]. Fig 6 shows that these transcripts occur predominantly in the same population as BARP, indicating that they are mainly expressed by epimastigotes. GCS1/HAP2 is also detected in the BARP-positive population; interestingly, cells expressing GCS1/ HAP2 are scattered throughout this population, rather than clustering as a separate cell type, as might be expected for gametes. Fig 6 also illustrates

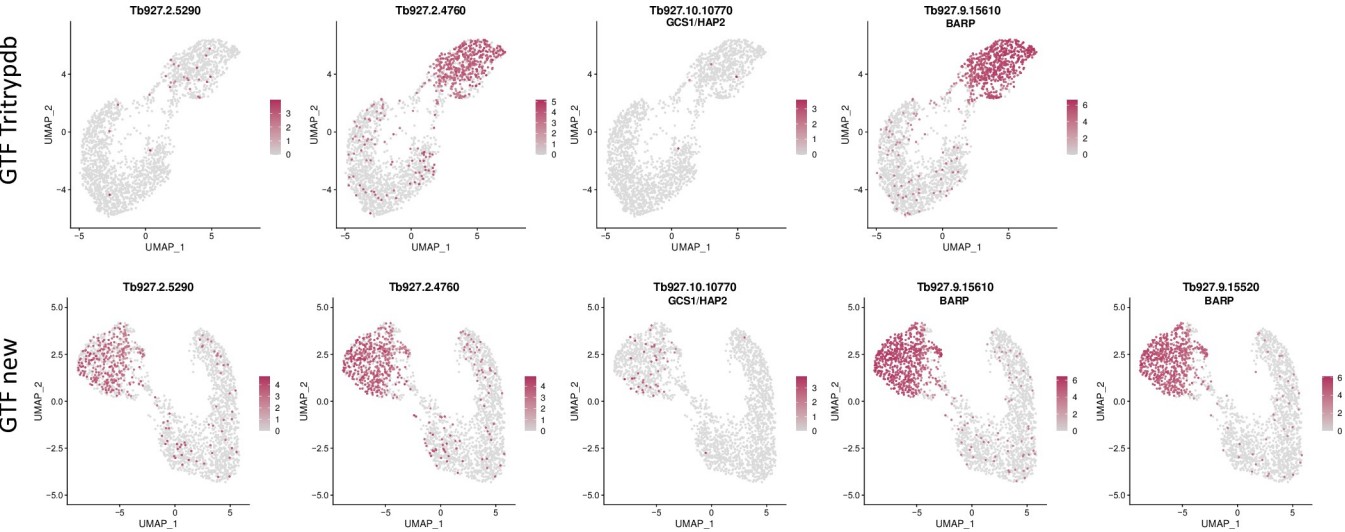

**Fig 6.** UMAP coloured by transcript counts for selected genes using GTF files from TriTrypDB (upper panel) and this study (GTF new, lower panel). The scRNA-seq dataset was obtained from Vigneron et al. [41]. The colour scale shows raw transcript counts per cell. The epimastigote cluster is identified by the presence of BARP transcripts. Different BARP isoforms are expressed to different extents. Tb927.9.15620 was not identified with the GTF file from TritrypDB.

that improved annotation of the GTF file led to the identification of transcripts that were completely missing from the GTF generated from TriTrypDB, for example BARP isotype Tb927.9.15620. Other transcripts were detected more frequently when the new version was used, such as Tb927.2.5290 and GCS1/HAP2.

Finally, we compared the fly transcriptome with the in vitro RBP6-overexpression data recently published by Doleželová and coworkers [70]. When we performed a principal component analysis, the two datasets showed no overlap (S2 Fig). This is not entirely surprising, however: first, in culture there is no separation of life-cycle stages that occur in separate tissues in the fly and second, the cells are cultured in a single medium, whereas trypanosomes in the fly will encounter different environments during their life cycle.

## Changes in adenylate cyclases, amino acid transporters and metabolic components during life cycle progression

The adenylate cyclase family is greatly expanded in *T. brucei* compared to other trypanosomatids [71]. This has been attributed to the complexity of its life cycle and the need for the parasites to sense their environment [61,71,72]. To cluster changes, a heatmap was generated for 82 adenylate cyclase genes (Fig 7). It was shown recently that cyclic AMP signalling is required for trypanosomes to move from the midgut lumen to the ectoperitrophic space [73]. Changes in expression of adenylate cyclases in the midgut are apparent throughout infection from D3 to D28. The most striking differences in expression occur between slender and stumpy bloodstream forms in the mammal, and midgut, proventriculus and salivary gland forms harvested on D28. Relative to the midgut on D28, 19 adenylate cyclases are upregulated and 24 are downregulated ≥2-fold in the proventriculus (S4 Table). Three adenylate cyclases previously characterised by Saada and coworkers in procyclic forms, ACP3, 4 and 5 [74], are downregulated >5-fold, while ACP2 is upregulated 3.2-fold (S3 and S4 Tables). Tb927.4.3870 and 4.3880 show the highest levels of upregulation (7-fold compared to midgut D28).

Amino acid transporters constitute another multigene family that shows extensive changes in expression through the life cycle (Fig 8). Once again, the largest changes are between the proventriculus and salivary glands on D28, with ≥2-fold upregulation of 9 amino acid transporters and downregulation of 5 others (S4 and S5 Tables), but there are also changes in the midgut over time. Closely related genes can be differentially regulated. For example, 3 members of the AAT7-B proline/alanine transporter family [17] are expressed at different times in the midgut (D3 or D15), one is biphasic (midgut D7 and proventriculus) and others show maximum expression in the proventriculus or in the salivary glands.

To obtain a broader picture of the changes occurring through the life cycle, a heatmap was generated for 61 enzymes involved in energy metabolism (Fig 9 and S2 Table). The clustered changes confirmed that glycolytic enzymes are upregulated at the beginning of infection and that enzymes of the TCA cycle are most highly expressed in the proventriculus, as well as identifying more subtle differences in individual enzymes. Data from the different analyses, together with localisation data from publications, TritrypDB and tryptag.org, were combined to create the overview in Fig 10. This suggests that the protein compositions of the glycosome and the mitochondrion change over time as the trypanosome moves from a reliance on glycolysis to proline catabolism. There is additional evidence for reduced dependence on glycolysis in the proventriculus. Fructose 2,6-bisphosphate is an allosteric activator of pyruvate kinase, the last enzyme in the pathway [75]. Proventricular forms express two isoforms of PFK-2/FBPase-2 (Tb927.10.4520 and Tb927.8.1020, formerly known as Tb3 and Tb4) [75,76]. The former has FBPase-2 activity, which converts fructose 2,6-bisphosphate to fructose-6-phosphate. Although proventricular forms express their own subset of proline/alanine transporters

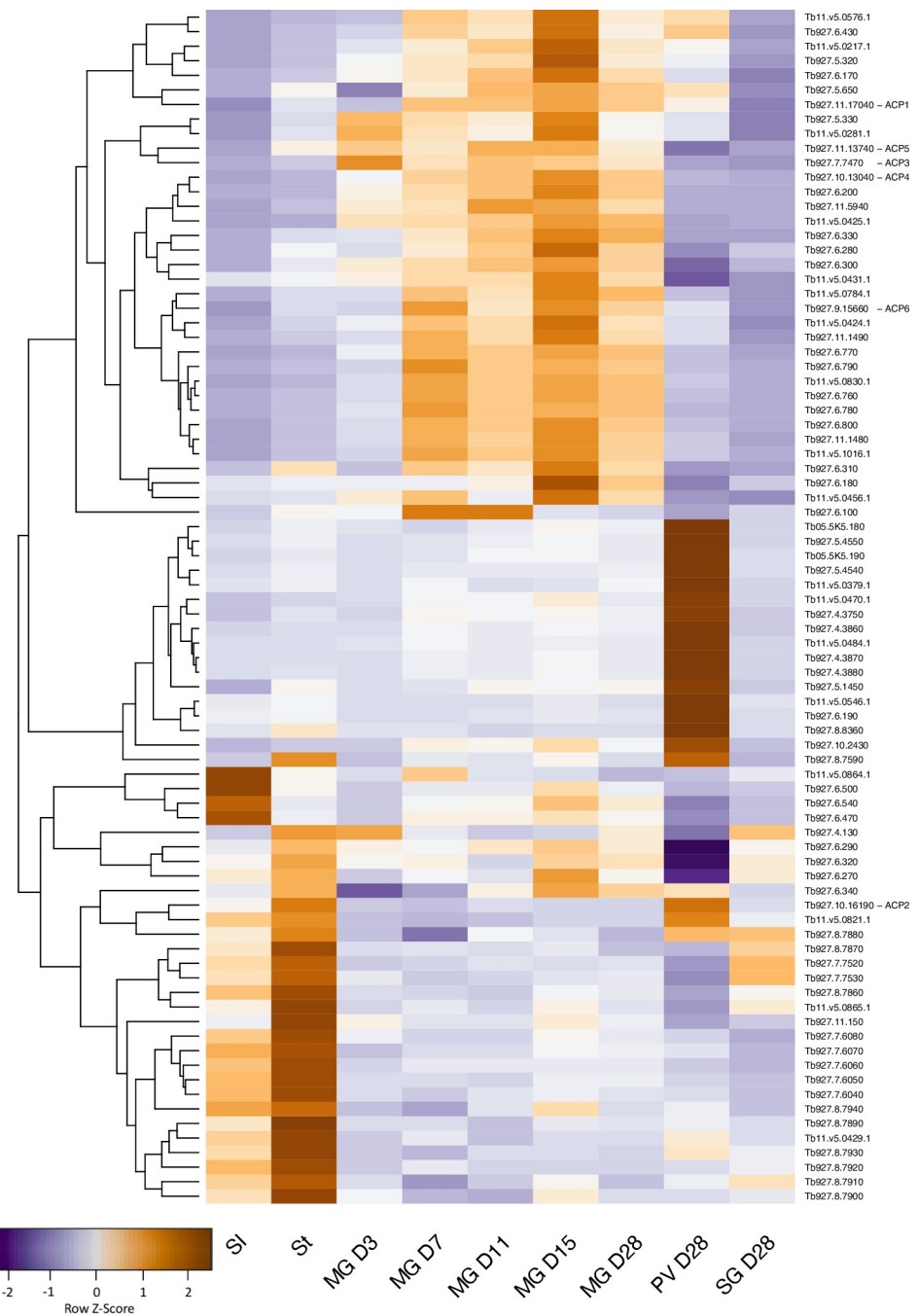

**Fig 7. Gene expression heatmap of adenylate cyclases, with pseudogenes excluded.** Rows indicate each gene and columns indicate samples. Data were z-score normalised. Colour scale: orange, high expression; blue, low expression.

it is possible that they are synthesising proline as well, since cytoplasmic delta-1-pyrroline-5-carboxylate reductase is upregulated [77]. The complexity of the population in the salivary glands does not allow us to attribute metabolic pathways to specific life-cycle stages. It does appear, however, that the TCA cycle no longer plays a major role. Moreover, transcripts for glycolytic enzymes such as pyruvate kinase and pyruvate dehydrogenase increase, as would be expected for metacyclic forms [43].

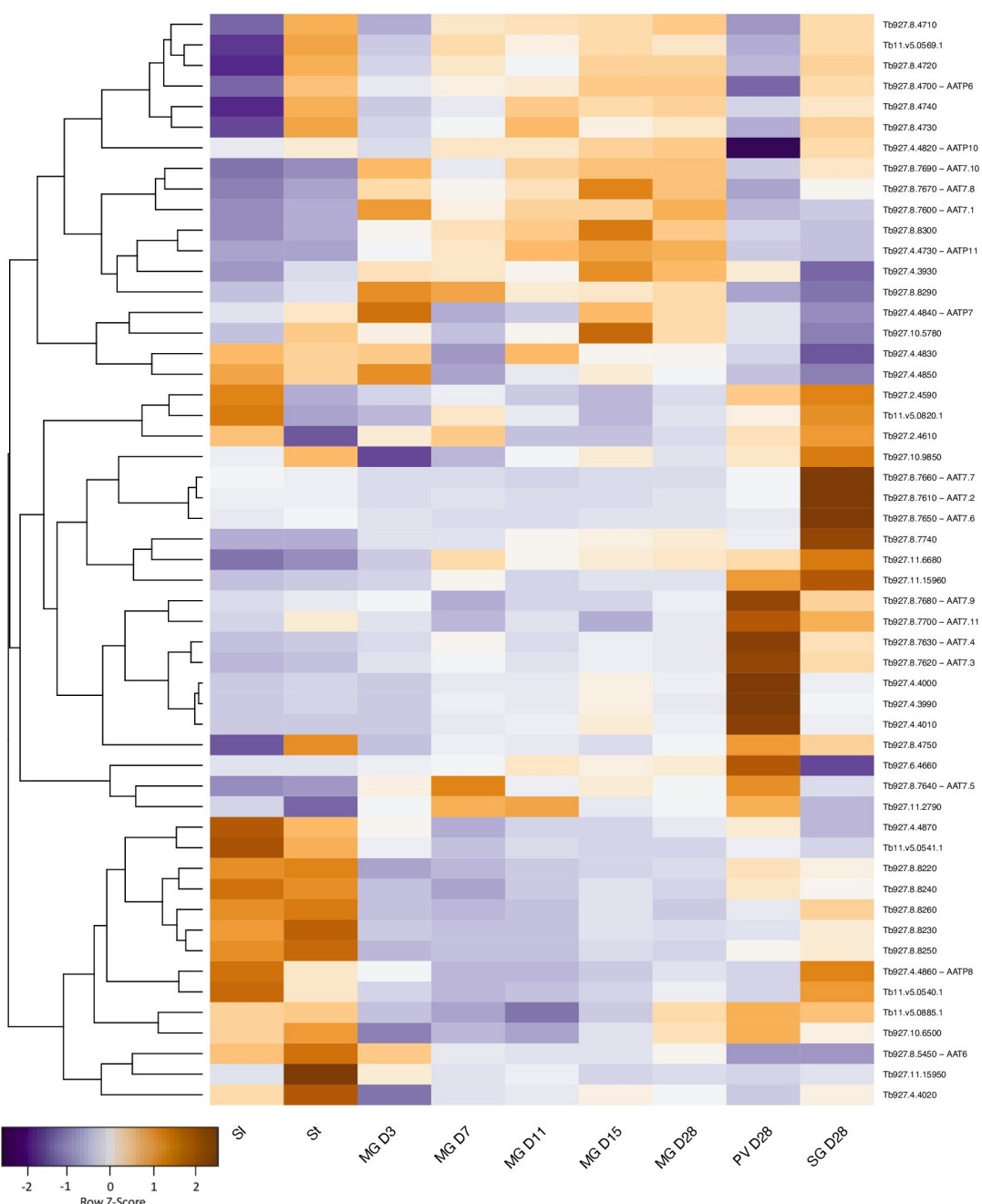

**Fig 8. Gene expression heatmap of amino acid transporters.** Rows indicate each gene and columns indicate samples. Data were z-score normalised. Colour scale: orange, high expression; blue, low expression. Members of the AAT7-B family are marked by asterisks.

## Discussion

Assembling the genome of *T. b. brucei* EATRO1125 and analysing its transcriptome has provided new information on how trypanosomes establish an infection in tsetse and progress through their life cycle. In addition to revealing that trypanosomes are metabolically more dynamic than was previously supposed, it has also led us to question accepted dogma. For example, it is often stated that there is no glucose in the tsetse midgut, other than shortly after

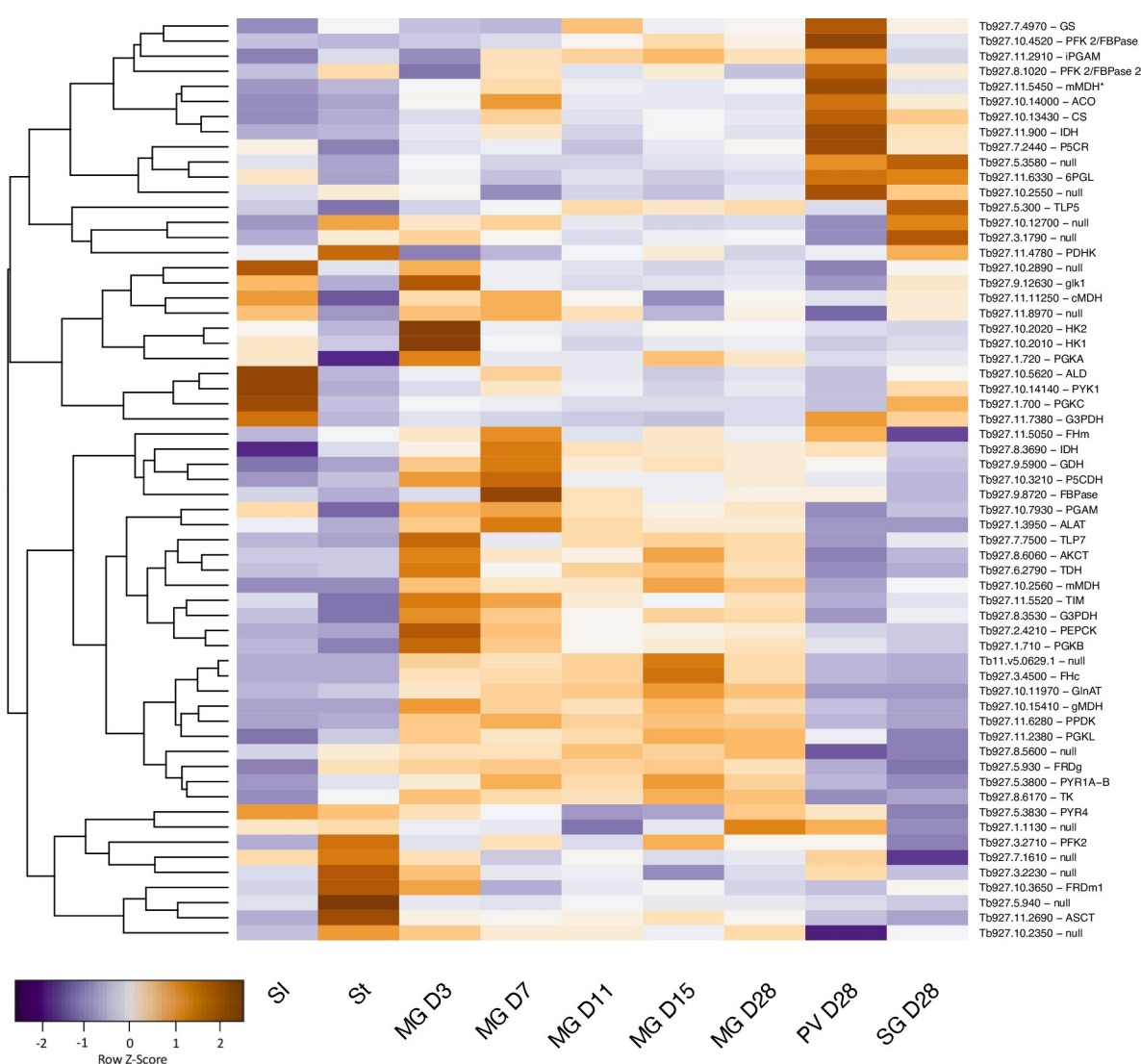

**Fig 9. Gene expression heatmap of selected metabolic enzymes.** Rows indicate each gene and columns indicate samples. Data were z-score normalised. Colour scale: orange, high expression; blue, low expression. Gene IDs and names are shown. TritrydDB uses the gene name "null" for some genes with putative functions that have not been confirmed experimentally.

a blood meal, but there are no published measurements. Our data on the transcriptome strongly suggests that midgut trypanosomes are able to acquire and metabolise glucose, at least for the first week of infection and possibly beyond (Fig 10). It is also possible that the parasites metabolise trehalose, a disaccharide that has been detected in tsetse [78]; *T. b. brucei* encodes a putative trehalose phosphorylase (Tb927.11.10670) that could generate glucose and glucose-1-phosphate. The ability to utilise glucose would provide a window for mitochondrial maturation and the switch to proline uptake and catabolism, which peaks on D7. Concomitant with this switch in energy source, transcripts for high affinity (THT2) glucose transporters are reduced and several amino acid transporters, including proline/alanine transporters of the AAT7-B family, are increased (Fig 10). Transient upregulation of three components of the TCA cycle also occurs on D7 and may coincide with the movement of the parasites into the anterior part of the ectoperitrophic space, close to the proventriculus. Although it was

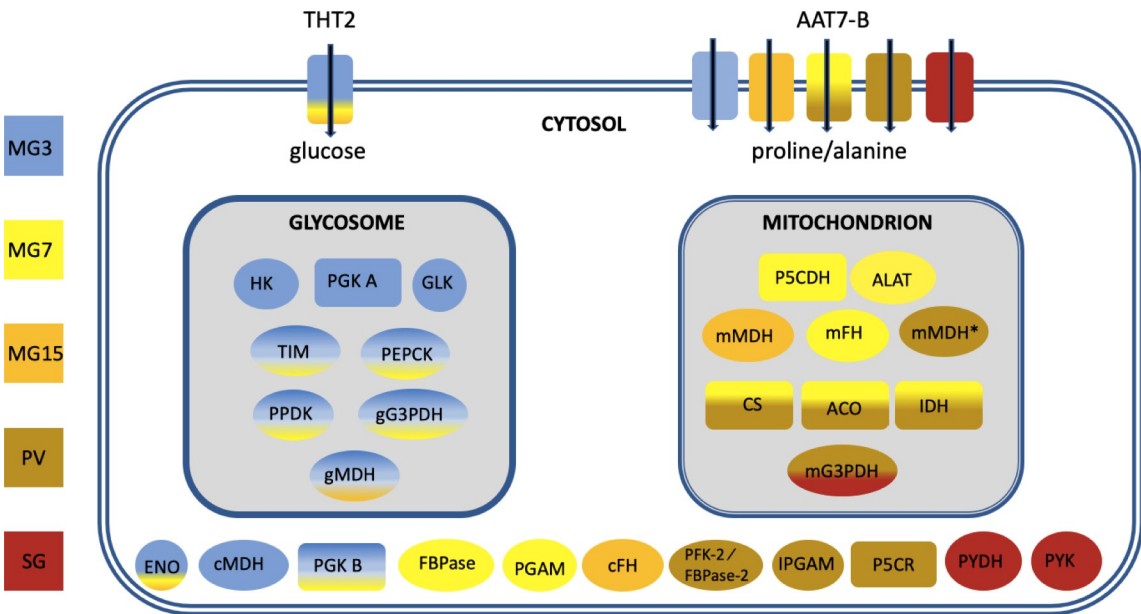

**Fig 10. Overview of changes in transcripts encoding metabolic enzymes during fly infection.** Gene IDs are provided in Fig 8. Colours reflect the time point or tissue in which expression was maximal. It does not imply that an enzyme is absent in other stages. AAT7-B: amino acid transporters; ACO: aconitase; ALAT: alanine aminotransferase; CS: citrate synthase; ENO: enolase; FBPase: fructose 1,6 biphosphatase; PFK-2/FBPase-2: fructose 2,6 bisphosphatase; cFH: cytosolic fumarate hydratase; mFH: mitochondrial fumarate hydratase GLK: glycerol kinase 1; gG3PDH: glycosomal glycerol-3-phosphase dehydrogenase; mG3PDH: mitochondrial glycerol-3-phosphase dehydrogenase; HK: hexokinase 1; IDH: isocitrate dehydrogenase; IPGAM: cofactor independent phosphoglycerate mutase; cMDH: cytosolic malate dehydrogenase; gMDH: glycosomal malate dehydrogenase; mMDH: mitochondrial malate dehydrogenase; mMDH*: mitochondrial malate dehydrogenase; P5CDH: pyrroline-5-carboxylate dehydrogenase; P5CR: pyrroline-5-carboxylate reductase; PEPCK: phosphoenolpyruvate carboxykinase; PGAM: phosphoglycerate mutase; PGK: phosphoglycerate kinase; PPDK: pyruvate phosphate dikinase; PYDH: pyruvate dehydrogenase; PYK: pyruvate kinase; TIM: triose phosphate isomerase; THT2: high affinity hexose transporters. MG3: midgut day 3; MG7: midgut day 7; MG15: midgut day 15; PV: proventriculus, SG: salivary glands.

previously reported that only part of the TCA cycle is functional in procyclic forms [79,80], a recent publication provided evidence for the full cycle [16]. It is likely that both sets of results are correct and that it depends on whether the cells studied were early or late procyclic culture forms or a mixture of the two. Unfortunately, this was not determined in either case. Markers identified for early and late procyclic culture forms are differentially expressed in the tsetse midgut, with markers for early forms predominating on D3. Hence, these two forms may represent the ends of a spectrum of procyclic forms with different metabolic capabilities. The differential expression of adenylate cyclases, several of which have been localised to the parasite flagellum [74], and amino acid transporters, further suggests that the midgut forms sense and react to different stimuli. Some of these changes may occur during the development of mesocyclic forms, which do not occur in culture. Taken together, the picture that emerges is one in which trypanosomes are constantly fine-tuning their response to their microenvironment.

There are two hypotheses about how the proventriculus is populated. One hypothesis is that trypanosomes are only capable of invading this organ during a defined window about 8–11 days post infection; if this does not occur, there will be no subsequent colonisation of the salivary glands [54]. Another hypothesis is that the proventriculus is not colonised permanently, but rather that it is constantly replenished by trypanosomes from the ectoperitrophic space [9]. Interestingly, some features of the midgut transcriptome at D7, such as upregulation of citrate synthase, aconitase and isocitrate dehydrogenase mRNAs are also seen in the proventriculus (Figs 4 and 10), raising the possibility that part of the midgut population is competent

to cross the peritrophic matrix into the proventriculus at this point. On the whole, however, proventricular forms have a transcriptome that differs markedly from that of midgut forms. Some of the most striking features are the differential expression of 43 adenylate cyclases and 14 amino acid transporters compared to the midgut on the same day. The only other life-cycle stage showing such large differences in adenylate cyclase expression is the stumpy form (Fig 7).

Metacyclic forms isolated from a single fly express different mVSGs [81,82]. Based on the ability of a pool of monoclonal antibodies to prevent infection, it was proposed that there were maximally 27 distinct variants, but it was not tested if multiple antibodies recognised the same mVSG [83]. Similar experiments with *T. congolense* provided an upper estimate of 12 mVSGs [84]. Our results suggest that the mVSG repertoire of EATRO1125 is smaller, as the same 7 mVSGs were detected in mRNA derived from 60 infected flies. Of these, 2 variants accounted for 61–69% of VSG transcripts in biological replicates and they remained the dominant mVSGs after fly passage, isolation of bloodstream forms and infection of new flies. The mVSG repertoire is not completely stable, however, and different laboratory stocks from the same original isolate may vary to some extent [85].

In addition to mVSGs, several other GPI-anchored proteins and a new family of membrane proteins with potential for mucin-like modifications are upregulated specifically in the salivary glands. The latter might be involved in adhesion to host epithelia or parasite-parasite interactions including gamete fusion. Related proteins are encoded by *T. congolense*, *T. vivax* and *T. cruzi*, but these have smaller stretches of threonine, if any. A comparison with RNA-seq data from RBP6-induced metacyclic culture forms [43] and single cell RNA-seq from salivary gland forms [41] indicates that, apart from mVSGs, neither the surface protein transcripts described above nor RHS7 belong to the metacyclic transcriptome. GCS1/HAP2 is a marker for male gametes in a variety of organisms, but there are no known common markers for female gametes. In *Plasmodium*, GPI-anchored proteins expressed by female gametes are required for recognition and fertilisation [86–88] and it is possible that some of the proteins up-regulated in the tsetse salivary glands will play a similar role in *T. brucei*.

Single cell sequencing of trypanosomes is still in its infancy [41,89], but it will ultimately be of great value in creating a cell atlas of life-cycle stages as trypanosomes progress through their hosts. One of the challenges, at present, has been the relatively short sequences obtained by single cell RNA technologies and the difficulties of mapping them correctly. To date, missing 3' UTRs have been dealt with empirically, either by extending a gene by a fixed length or extending it down to the next coding sequence. The annotated file provided here improves the accuracy of mapping, particularly of transcripts that are not expressed in bloodstream and procyclic forms. Furthermore, although the genomes of different isolates of *T. b. brucei* are very similar to each other, there are single nucleotide polymorphisms and indels that could influence the design of guide RNAs for genome editing. In this regard, the genome assembly of EATRO1125 will provide a useful resource for the community.

## Supporting information

**S1 Fig. Schematic drawing of tsetse fly alimentary tract.**
(TIF)

**S2 Fig. Principal component analysis of trypanosome transcriptomes from different tsetse tissues (this study) and RBP6-overexpressing cells (70).**
(TIF)

**S1 Table. Genomic sequences, reads per contig.** Contigs with telomeres and/or 177bp repeats are annotated.
(XLSX)

**S2 Table. Mean expression values (reads per million) and raw read counts for all transcripts.** Sl: slender bloodstream forms; St: stumpy bloodstream forms; MG3: midgut day 3; MG7: midgut day 7; MG15: midgut day 15; PV: proventriculus, SG: salivary glands. Three biological replicates were performed for each tsetse-derived sample.
(XLSX)

**S3 Table. Pairwise comparison of differentially expressed genes in midgut forms on different days post infection.**
(XLSX)

**S4 Table. Genes differentially expressed between midgut and proventricular forms on day 28 post infection.**
(XLSX)

**S5 Table. Genes differentially expressed between proventricular and salivary gland forms on day 28 post infection.**
(XLSX)

**S1 File. Alignment of EATRO1125 contigs to TREU927.**
(PPTX)

**S2 File. GFF3 file.**
(GFF3)

**S3 File. Transcript assemblies.**
(GZ)

**S4 File. GTF.**
(GTF)

**S5 File. Transcript assemblies of metacyclic VSG sequences.**
(DOCX)

## Acknowledgments

We are grateful to Nina Papavasiliou and the Genomics and Proteomics Core Facility of the German Cancer Centre, Heidelberg for a subset of PacBio sequences, and Galaxy Main (usegalaxy.org), Galaxy Australia (usegalaxy.org.au) and the UBELIX HPC cluster at the University of Bern (www.id.unibe.ch/hpc) for providing computing facilities. Berta Pozzi and Ruth Etzensperger are thanked for helpful comments on the manuscript and Manon Vonlaufen is thanked for drawing S1 Fig.

## Author Contributions

**Conceptualization:** Arunasalam Naguleswaran, Isabel Roditi.

**Data curation:** Arunasalam Naguleswaran, Pamela Nicholson, Isabel Roditi.

**Formal analysis:** Arunasalam Naguleswaran, Paula Fernandes, Pamela Nicholson, Isabel Roditi.

**Funding acquisition:** Isabel Roditi.

**Investigation:** Arunasalam Naguleswaran, Isabel Roditi.

**Methodology:** Arunasalam Naguleswaran, Paula Fernandes, Shubha Bevkal, Ruth Rehmann, Pamela Nicholson, Isabel Roditi.

**Project administration:** Isabel Roditi.

**Supervision:** Isabel Roditi.

**Visualization:** Arunasalam Naguleswaran, Paula Fernandes.

**Writing – original draft:** Arunasalam Naguleswaran, Isabel Roditi.

**Writing – review & editing:** Arunasalam Naguleswaran, Paula Fernandes, Shubha Bevkal, Pamela Nicholson, Isabel Roditi.

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
