## [Decision Letter · Decision Letter 0]

7 Jun 2021

Dear Prof. Roditi,

Thank you very much for submitting your manuscript "Developmental changes and metabolic reprogramming during establishment of infection and progression of Trypanosoma brucei brucei through its insect host" for consideration at PLOS Neglected Tropical Diseases. As with all papers reviewed by the journal, your manuscript was reviewed by members of the editorial board and by several independent reviewers. In light of the reviews (below this email), we would like to invite the resubmission of a significantly-revised version that takes into account the reviewers' comments. 

We cannot make any decision about publication until we have seen the revised manuscript and your response to the reviewers' comments. Your revised manuscript is also likely to be sent to reviewers for further evaluation.

Sincerely,

Daniel K. Masiga

Deputy Editor

Daniel Masiga

Deputy Editor

Reviewer's Responses to Questions

**Key Review Criteria Required for Acceptance?**

**Methods**

-Are the objectives of the study clearly articulated with a clear testable hypothesis stated?

-Is the study design appropriate to address the stated objectives?

-Is the population clearly described and appropriate for the hypothesis being tested?

-Is the sample size sufficient to ensure adequate power to address the hypothesis being tested?

-Were correct statistical analysis used to support conclusions?

-Are there concerns about ethical or regulatory requirements being met?

Reviewer #1: A Naguleswaran and colleagues present a genome assembly of Trypanosoma brucei brucei EATRO1125 (AnTaR1 serodeme), an isolate that is used increasingly often because it is pleomorphic in mammals and fly transmissible, two characteristics that have been lost by the most commonly used laboratory stocks. They also performed de novo transcriptome assemblies using Illumina sequences, with reads long enough to discriminate related members of multigene families, from tsetse-derived trypanosomes isolated from the midgut, the proventriculus and the salivary glands of infected flies. 

The methods used are state-of-the-art genomics and transcriptomics approaches.

Reviewer #2: See assembled review comments

Reviewer #3: Yes.

**Results**

-Does the analysis presented match the analysis plan?

-Are the results clearly and completely presented?

-Are the figures (Tables, Images) of sufficient quality for clarity?

Reviewer #1: This revealed that the transcriptome of midgut-derived parasites is dynamic and resemble the transition from early to late procyclic forms in culture. In addition, RNA-Seq data provides relevant informations in the metabolic reprogramming occurring the proventriculus, including upregulation of tricarboxylic acid cycle enzymes. Many transcripts upregulated in the salivary glands encode surface proteins, among them 7 metacyclic VSGs, multiple BARPs, a marker for gametes (GCS1/HAP2) and a novel family of transmembrane proteins containing polythreonine stretches.

The data are clearly presented and the figures are well designed.

Reviewer #2: See assembled review comments

Reviewer #3: Yes.

**Conclusions**

-Are the conclusions supported by the data presented?

-Are the limitations of analysis clearly described?

-Do the authors discuss how these data can be helpful to advance our understanding of the topic under study?

-Is public health relevance addressed?

Reviewer #1: This combination of genomic (DNA) and transcriptomic (RNA) data provides new insights into trypanosome biology and will be useful resources for the trypanosome research community, including identification of transcripts obtained by single cell sequencing technologies.

Reviewer #2: See assembled review comments

Reviewer #3: Yes.

**Editorial and Data Presentation Modifications?**

Reviewer #1: I only have a few minor points that could be addressed by the authors.

1. Lanes 449-451. The PFK-2 (6-phosphofructo-2-kinase)/FBPase-2 (fructose-2,6-bisphosphatase) enzyme produces or dephosphorylates fructose 2,6-bisphosphate (Fru-2,6-P2), a signal molecule that controls glycolysis and gluconeogenesis (Rider et al, 2004, Biochem J). Consequently, PFK-2/FBPase-2 is not involved in the production of the glycolytic intermediate fructose 1,6-bisphosphate (Fru-1,6-P2). In trypanosomes, Fru-2,6-P2 has been described as the main allosteric activator of pyruvate kinase and is involved in the control of the metabolic flux, rather than being a metabolic intermediate of glycolysis/gluconeogenesis. It seems that the authors have confused PFK-2/FBPase-2, and PFK and FBPase. This should be clarified in the text.

2. Lane 414. "Table S4" should be replaced by "Table S5"

3. Figures 6 and 7. It would be relevant and very convenient for the readers to add, after the gene number, the name of the gene when known, as well as the function, in particular for amino acid carriers, as the author did for Figure 8.

Reviewer #2: See assembled review comments

Reviewer #3: Minor revision.

**Summary and General Comments**

Reviewer #1: This combination of genomic (DNA) and transcriptomic (RNA) data provides new insights into trypanosome biology and will be useful resources for the trypanosome research community, including identification of transcripts obtained by single cell sequencing technologies.

Reviewer #2: This manuscript combines two components. Firstly, it provides a genome resource for T. brucei EATRO 11125 (‘AnTat’s’) that will be of value to the trypanosome research community that focus on developmental events and the in vivo biology of the parasite. Secondly, it provides a transcriptome description of several developmental stages as the parasite progresses through the tsetse fly, adding to existing datasets derived from salivary gland parasites, as well as very detailed transcriptome and proteomic studies of parasite development induced artificially in culture through RBP6 overexpression. Again, this provides a useful resource for the research community.

Overall, the datasets are of value and will be valuable when incorporated into TriTryp DB where they can add to the information available for researchers when understanding gene function in the parasite. I will break my comments into those concerned with the genome analysis and those concerned with the transcriptome analyses. 

Note that these quite extensive notes and requests for clarification or technical amendment are to improve the utility and accessibility of the data and to reassure readers and provide maximal value to them. Overall, I consider the data provided will be a valuable resource that should be published. 

1. Genome analysis

Although their methodology is likely to have produced a usable genome assembly, especially due to the use of the companion pseudochromosome assignment tool, I would like to see a few more quality control and assembly details before it is published. For example;

• I was unclear how much data was produced with the PacBio DNA run? What was the predicted genome coverage?

• It doesn’t appear error correction for the Pacbio reads was carried out by illumina- is this the case?

• Some basic assembly statistics should be provided (e.g., N50, contig number, GC content, numN50, mean contig size, max and min length of contigs) and this should be compared to a reference genome (e.g., Tb927).

• A BUSCO assessment is needed for the genome assembly – this provides an unbiased assessment of the assembly - completed, fragmented and/or duplicated.

• The coverage of each contig should be provided. I realise the number of reads mapping to each contig is provided, but this doesn’t summarise the coverage of each contig. 842 contigs seems to be very high and a quick examination of Table S1 shows there seems to be several contigs with very few reads mapping (often only 1 read!). I wonder if the assembly should be collapsed to remove duplication?

• Although cultured cells were used for raw genome data, levels of contaminating DNA are best assessed via BLAST searches of each contig in case unrelated DNA has crept in during isolation/library preparation and therefore assembly. 

• More detail could be provided when describing the assignment of contigs to chromosomes and for those resulting contigs which are suggested to derive from mini/intermediate chromosomes. For example, companion does assign contigs to the 11 megabase chromosomes but it is unclear which contigs have been mapped to these 11 chromosomes. It is stated that “The remaining contigs (28.1 Mb in total) could not be assigned to the megabase chromosomes. Of these, 73 contigs contained 177 bp repeats, which are characteristic of mini- and intermediate chromosomes (38), and 98 contained telomeric repeats. Many of these contigs were close to the expected sizes of mini- and intermediate chromosomes. Further analysis revealed that the majority of them encoded a single VSG at the end of the contig; few encoded more than one VSG.” 

Based on this some further clarity is needed:

How many contigs were remaining after the assignment to megabase chromosomes?

Please quantify the statement about many contigs being of the expected mini/intermediate chromosome size (size and number of contigs).

• It would be valuable to provide the alignment of contigs derived from AnTat to Tb927. This figure should be produced by default during the companion pipeline and this would show any genomic regions with potential deletions/ expansions in Antat, which is valuable information.

• The authors go onto validate their assembly by looking at known regions. They note that “In addition, the assembly of chromosome 10 confirmed the previously described organisation of EP1 and EP2 procyclin and downstream procyclin associated genes (PAGs) 1, 2, 4 and 5 on one copy of the chromosome and a PAG1/2 fusion, and no PAG5 on the second copy (40).” 

Are both copies of chromosome 10 included- I understood this was a haploid assembly? 

BUSCO could provide a more accurate and unbiased approach to this this assessment. 

• a k-mer based assessment of their raw data could be run to estimate the predicted genome size before assembly. This would give a good idea whether duplication in the assembly is present, or if it is simply the mini/ intermediate chromosomes that have almost doubled the size of the assembly in comparison to Tb927.

• They could also run purge_dups (https://github.com/dfguan/purge_dups) to remove potential duplication caused by heterozygous regions of the genome.

2. Annotation 

• Annotation: What settings were used with companion? There are many settings to choose from and the setting used should be stated. For example, in ‘Step 6: Advanced settings of companion’: changing the sensitivity of Augustus will drastically alter the number of genes annotated. Please clarify.

• Was the transcriptome data used during the companion annotation run?

• By running BUSCO on the companion protein file, in protein mode, it would also be possible to show how successful the annotation was. 

3. Transcriptome assembly 

Although the use of the 927 genome for transcriptome assignment was justified to allow comparison with earlier datasets, this seems a strange decision given the effort put in to deriving the AnTat genome. Further, the manuscript does not directly compare the current datasets with results from other labs using 927 except in the case of some specific gene sets- although I think this would be useful (see below).

I wonder what was the overall mapping rate of RNA reads to the new Antat pseudo-assembly?

4. RNA seq analysis

The dataset generated is valuable and provides a good resource to explore transcripts expressed by the parasites as they traverse compartments within the tsetse fly. It is largely descriptive however and I feel an opportunity has been missed to compare the in vivo data with the impressive and high-quality transcriptome, proteome (and metabolome) datasets generated by the Zikova lab recently using RBP6 overexpression (Doleželová et al, PLOS Biology 2020; https://doi.org/10.1371/journal.pbio.3000741). The extent to which the datasets correlate would be very valuable to the community and allow better assessment of the utility of the in vitro model which is now being used extensively by several laboratories. Similarly direct comparison with the single cell data (Vigneron et al, PNAS 2020) and RNA seq data from infected tsetse salivary glands (Savage et al., PLOS One 2016) would be valuable.

Currently the figures summarising the expression profiles and datasets are quite limited, comprising heatmaps for different gene families and a non-quantitative summary figure of metabolism. I would hope a more detailed and intuitive analysis could be presented of the respective comparison of the current datasets with other datasets at different timepoints or with different experimental systems. This should be feasible via PCA plots for example to provide an assessment of the level of correspondence between, for example, published in vitro and in vivo models for some developmental stages in the fly, or single cell RNA data for the salivary gland and the RNAseq data provided here.

The details of the changes of various transcripts were often quite descriptive and transcripts highlighted without a clear indication of why these had been selected for description. For example, ““Several transcripts for metabolic enzymes peaked on D7, including fructose-1,6-bisphosphatase and the tricarboxylic acid (TCA) cycle enzymes citrate synthase, isocitrate dehydrogenase and aconitase (Figure 4; Table S3). Transcripts for a proline/alanine transporter from the AAT7-B family (Tb927.8.7640) (17) and delta-1-pyrroline-5-carboxylate dehydrogenase also peaked on D7.” Were these changes significant? What selection criteria were used for highlighting changes?

Table 1: please add LogFC and significance to the Table

With respect to Figures 6,7 and 8. These figures are missing a scale for the colours used in the heatmap. Also have the genes and samples been clustered during plotting?

Some technical points that would be good to clarify with respect to the transcriptome analysis

• I found the methods for the transcriptome section slightly outdated. They deviate from the methods suggested by the DEseq2 vignette - http://www.bioconductor.org/packages/release/bioc/vignettes/DESeq2/inst/doc/DESeq2.html#differential-expression-analysis

• Please confirm the type of data that was passed to DESeq2 – raw or RPM values? – Deseq2 requires untransformed data

• Which version of Trinity was used for the de novo transcriptome assembly?

• A BUSCO completeness assessment should be performed on the transcriptome assembly.

• “When we compared this annotated file with the single cell RNA sequencing data generated by Vigneron and co-workers (41), we found an excellent correlation between their 3' reads and the mapped ends of transcripts” . How is an excellent correlation defined in this context?

• RPM is used for comparisons of transcript levels, but this does not account for gene length. TPM is more suitable I believe.

• “Moreover, the percentage of reads assigned confidently to the transcriptome increased from 41% to 48.5%”. Please specify what is meant by a confident assignment. Is a mapping quality cut off used? I don’t believe this was specified in the methods section.

Reviewer #3: This is a descriptive but very useful article describing (i) the genome of a pleomorphic strain of Trypanosoma brucei that a growing number of labs is using (EATRO1125), and (ii) the transcriptome of parasites colonising different organs of the Tsetse fly (the insect host). Having the genome of EATRO1125 will increase the accuracy of future genome-wide analysis and it will improve the efficiency of editing using CRIPSR/CAS9 technologies. The comparison of the transcriptome of in vivo parasites from midgut (several time points), proventriculus and salivary gland showed that there is a constant remodeling of gene expression, including of metabolic proteins, surface proteins and sensing factors (such adenylate cyclases). The transcriptome studies also allowed the identification of ~9300 genes, in which the 5’ and 3’UTRS were mapped, which is another useful tool for the community.

The experimental work is solid and the conclusions are sustained by the results obtained. I have a few minor suggestions, most of which aim to improve the clarity of figures and supplementary materials.

Minor comments

1. The pairwise comparisons described in Figure 3 and Table S3 are useful to identify the genes differentially expressed between two stages. However, an additional figure is missing to show the global patterns of transcript changes: to visually see which stage is more different, how many genes contribute to that difference, the level of up/down regulation. This could be achieved with a heat-map of all genes that are differentially expressed in at least one of the 7 time-points collected from the tsetse (similar to the heat-map of Figures 6-8); along with this global analysis, the authors could ask if there are co-regulated genes with similar patterns of gene expression in some or all stages in the Tsetse. This co-expression may result from a common regulatory mechanism, which could depend on conserved elements in 5’ or 3’UTR (if they exist). These global analysis could actually be part of Figure 1, after the PCA plot.

2. The paper should include a figure about the genome, showing a schematic of megabase, intermediate and mini-chromosomes, their sizes and numbers. Authors could also consider adding a table displaying some of the data described in the text, such as saying how many contigs, how many genes identified, etc.

3. For those readers less familiar with Tsetse life cycle stages, I suggest including a cartoon of a tsetse indicating the midgut, proventriculus and salivary glands. 

4. In data presented in Figure 3A/B, the authors should indicate the number of differentially expressed genes in the text and perhaps even in the top of each volcano plot. I know this information is available in Table S3, but indicating it in the figure would help the reader.

5. Line 414. I believe the data about RHS7 is in Table S5, not Table S4.

6. In T. brucei, VSG genes are classified as a- or b-type. To which type do metacyclic VSG identified in this study belong ? Is there any sequence similarity to metacyclic VSG genes in T. congolense? I suspect not.

7. Given that Table 1 depicts surface protein transcripts, shouldn’t GCS1/HAP2 be included?

8. In Figure 8, the authors should label the clusters of genes encoding for proteins involved in Glycolysis, TCA, etc. Otherwise, it is not really possible to visually interpret the heat map in terms of changes in metabolic functions.

9. Supplementary tables – Could the author add the title of the table as line 1? This will help go through the multiple supplementary files and more clearly identify each file.

PLOS authors have the option to publish the peer review history of their article (what does this mean?). If published, this will include your full peer review and any attached files.

Reviewer #1: No

Reviewer #2: No

Reviewer #3: No
---

## [Decision Letter · Decision Letter 1]

23 Aug 2021

Dear Prof. Roditi,

Thank you very much for submitting your manuscript "Developmental changes and metabolic reprogramming during establishment of infection and progression of Trypanosoma brucei brucei through its insect host" for consideration at PLOS Neglected Tropical Diseases. As with all papers reviewed by the journal, your manuscript was reviewed by members of the editorial board and by several independent reviewers. The reviewers appreciated the attention to an important topic. Based on the reviews, we are likely to accept this manuscript for publication, providing that you modify the manuscript according to the review recommendations. 

Sincerely,

Daniel K. Masiga

Deputy Editor

Daniel Masiga

Deputy Editor

Reviewer's Responses to Questions

**Key Review Criteria Required for Acceptance?**

**Methods**

-Are the objectives of the study clearly articulated with a clear testable hypothesis stated?

-Is the study design appropriate to address the stated objectives?

-Is the population clearly described and appropriate for the hypothesis being tested?

-Is the sample size sufficient to ensure adequate power to address the hypothesis being tested?

-Were correct statistical analysis used to support conclusions?

-Are there concerns about ethical or regulatory requirements being met?

Reviewer #1: (No Response)

Reviewer #2: see general comments

Reviewer #3: N/A

**Results**

-Does the analysis presented match the analysis plan?

-Are the results clearly and completely presented?

-Are the figures (Tables, Images) of sufficient quality for clarity?

Reviewer #1: (No Response)

Reviewer #2: see general comments

Reviewer #3: The authors have aptly addressed my previous concerns and questions in this revised manuscript.

**Conclusions**

-Are the conclusions supported by the data presented?

-Are the limitations of analysis clearly described?

-Do the authors discuss how these data can be helpful to advance our understanding of the topic under study?

-Is public health relevance addressed?

Reviewer #1: (No Response)

Reviewer #2: see general comments

Reviewer #3: The authors have aptly addressed my previous concerns and questions in this revised manuscript.

**Editorial and Data Presentation Modifications?**

Reviewer #1: (No Response)

Reviewer #2: see general comments

Reviewer #3: (No Response)

**Summary and General Comments**

Reviewer #1: From my point of view, the corrections made by the authors to the manuscript in response to the reviewer's comments are correct

Reviewer #2: The authors have done a good job at rebutting earlier reviewer comments, although a slightly a less good job of revising the manuscript to reflect these rebuttals. I suggest some of the comments in the rebuttal make it into the manuscript with, where appropriate, some discussion.

For example:

The BUSCO values are now included for the genome assembly and included in the Supplementary Table but there is no discussion of this in the main manuscript text. Please include comments in the main manuscript. Please also specify the version of BUSCO used and the database. Indeed, please go through the manuscript to ensure all analysis packages state the version used as this can have an important impact on their outputs and so the reproducibility of the study.

The coverage data is now included though significant numbers of contigs have a coverage of only 1 or 2, which greatly reduces confidence. Please include a comment in the main manuscript- not simply the Table values.

The newly included analysis of genome size indicates some expansion of the AnTat genome by around 3Mb. Please elaborate on the basis of this expansion in the manuscript text.

Comparisons with previously published datasets are now included via a PCA map in the rebuttal (which should be included as a supplementary figure). There is quite a large difference from, particularly, the datasets generated via RBP6 overexpression. Similarly the scRNA analysis (Figure 6 now includes this). Good justification for these differences is provided in the rebuttal but there is not an equivalent discussion in the main manuscript I think. I consider an expansion of text would be helpful to clarify why datasets between labs likely differ so that readers can best interpret the respective studies.

Reviewer #3: The authors have aptly addressed my previous concerns and questions in this revised manuscript. The analysis shown in the new Figure 6 has nicely improved the conclusions.

PLOS authors have the option to publish the peer review history of their article (what does this mean?). If published, this will include your full peer review and any attached files.

Reviewer #1: Yes: Frédéric BRINGAUD

Reviewer #2: No

Reviewer #3: No

Figure Files:

Data Requirements:

Reproducibility:

References

---

## [Editor Report · Decision Letter 2]

7 Sep 2021

Dear Prof. Roditi,

We are pleased to inform you that your manuscript 'Developmental changes and metabolic reprogramming during establishment of infection and progression of Trypanosoma brucei brucei through its insect host' has been provisionally accepted for publication in PLOS Neglected Tropical Diseases.

Best regards,

Daniel K. Masiga

Deputy Editor

Daniel Masiga

Deputy Editor

---

## [Editor Report · Acceptance letter]

14 Sep 2021

Dear Prof. Roditi,

We are delighted to inform you that your manuscript, "Developmental changes and metabolic reprogramming during establishment of infection and progression of Trypanosoma brucei brucei through its insect host," has been formally accepted for publication in PLOS Neglected Tropical Diseases.

Best regards,

Shaden Kamhawi

co-Editor-in-Chief

Paul Brindley

co-Editor-in-Chief
